

# The feasibility of virtual reality therapy for upper extremity mobilization during and after intensive care unit admission

Mirthe de Vries[1,2,*], Lise F.E. Beumeler[1,3,4,*], Johan van der Meulen[5], Carina Bethlehem[1], Rob den Otter[2] and E. Christiaan Boerma[1,3]

[1] Department of Intensive Care, Medical Center Leeuwarden, Leeuwarden, Netherlands
[2] Department of Human Movement Sciences, University of Groningen, Groningen, Netherlands
[3] Campus Fryslân, University of Groningen, Leeuwarden, Netherlands
[4] Research Group Digital Innovation in Healthcare and Social Work, NHL Stenden University of Applied Sciences, Leeuwarden, Netherlands
[5] 8D Games, Leeuwarden, Netherlands
[*] These authors contributed equally to this work.

## ABSTRACT

**Introduction**. Early mobilization reduces long-term muscle weakness after intensive care unit (ICU) admission, but barriers (*e.g.*, anxiety, lack of motivation) may complicate patients' adherence to exercise. Virtual reality (VR) presents immersive stimuli, which may increase motivation and adherence. This study aimed to examine the feasibility of VR-therapy using a VR-headset during ICU- and subsequent general ward admission. Furthermore, physical parameters before and after training were explored.

**Materials & Methods**. Ten adult ICU-patients with a median age of 71 [63–79], 70% of male registered birth sex, mechanically ventilated for $\geq$48 h, and willing to participate, were included. VR-therapy was offered three times a week for 20 minutes in addition to standard care. To train upper extremity functionality, patients were instructed to complete puzzles with increasing level of difficulty. Feasibility was based on patient satisfaction, session efficiency, and adherence levels during the training. Fatigue was measured after each session using the Borg Rating of Perceived Exertion Scale. Patients' hand-grip strength and Morton Mobility Index (MMI) were evaluated at the start of VR-therapy and after four weeks of training or at hospital discharge.

**Results**. On average, patients followed three VR-therapy sessions of 20 min per week with 13 min of actual training time, over the course of 1 to 3 weeks depending on their length of stay. Session efficiency ranged from 25% to 93%. In total, patients adhered to 60% of the VR-therapy sessions. MMI scores increased significantly from the start (26 [24–44]) to the end of the VR-therapy training period (57 [41–85], $p = 0.005$), indicating improved balance and mobility.

**Conclusion**. VR-therapy for upper extremity rehabilitation in ICU-patients is feasible during stay in the ICU and general ward.

Corresponding author
Lise F.E. Beumeler,
l.f.e.beumeler@rug.nl

## INTRODUCTION

In the Netherlands, an average of 80,000 patients are admitted to the intensive care unit (ICU) annually (*Stichting Nationale Intensive Care Evaluatie, NICE*). Reasons for an ICU-admission are diverse, including major operations, trauma and infection (*Wolters & Schuckman, 2021*). Advances in the management of critically ill patients have led to an increase in survival, but not necessarily to an improvement in quality of life (*Zimmerman, Kramer & Knaus, 2013*; *Rengel et al., 2019*). Many ICU-survivors suffer from newly developed or worsened long-term mental (*e.g.*, cognitive dysfunction, emotional distress) and physical impairments (*e.g.*, muscle weakness, reduced endurance) as a result of ICU-treatment (*Rengel et al., 2019*), termed Post Intensive Care Syndrome (PICS) (*Needham et al., 2012*). It is estimated that 50–70% of the ICU-survivors suffer from PICS one year after ICU-admission (*Tipping et al., 2017*; *Puthucheary et al., 2013*). The growing number of ICU-survivors with PICS shows the need to address long-term consequences more fully.

Muscle weakness, referred to as ICU-acquired weakness (ICU-AW), is one of the consequences of critical illness and immobilization. ICU-AW occurs within 24 h and continues to progress during admission (*Tipping et al., 2017*; *Puthucheary et al., 2013*; *Azoulay et al., 2017*). In the ICU, mobilization is therefore started as early as possible to diminish long-term muscle weakness. Early mobilization could include any combination of bed mobility practice, active exercises in bed, transfers from sitting to standing and walking, or lifting to a chair (*Tipping et al., 2017*). Early mobilization is feasible, safe, and can improve muscle strength and function at ICU-discharge (*Hodgson et al., 2016*; *Kayambu, Boots & Paratz, 2015*; *Dantas et al., 2012*; *Needham, 2008*; *Schweickert et al., 2009*). However, in clinical practice, there are barriers to implement early mobilization, such as lack of staff, equipment, and knowledge (*Parry et al., 2017*). Moreover, patient anxiety and lack of motivation, confidence, and knowledge about ICU-AW are identified as barriers impeding adherence to early mobilization (*Williams & Flynn, 2013*). The ideal early mobilization program should deliver therapy that is feasible for staff as well as safe and motivating for patients.

The use of exergames, or technology-driven physical activities, may provide a solution to address barriers of delivering early ICU mobilization in a fun, relaxed way. Previously, Virtual Therapy Environments using virtual platforms like Xbox Kinect Jintronix© software and the Nintendo Wii™ were successfully applied in the ICU setting (*Parke, Hough & Bunnell, 2020*; *Gomes, Schujmann & Fu, 2019*). More recently, the potential of immersive technology in the ICU, like virtual reality (VR), has been explored, with a primary focus in patient studies on relaxation and pain management (*Kanschik et al., 2023*). VR can influence patient behavior by presenting strong immersive stimuli and its ability to provide a feeling of presence and emotional engagement in a virtual three-dimensional world (*Elor & Kurniawan, 2020*; *Bohil, Alicea & Biocca, 2011*; *Tieri et al., 2018*). Exercises embedded in VR are more engaging than in a sterile medical setting, which may increase patient motivation and subsequent adherence to therapy (*Bohil, Alicea & Biocca, 2011*; *Tieri et al., 2018*; *Meekes & Stanmore, 2017*). VR is a helpful

tool to recover cognitive and motor functioning of populations with neurodegenerative diseases, traumatic brain injury, and stroke (*Bohil, Alicea & Biocca, 2011*; *Tieri et al., 2018*; *Rizzo & Koenig, 2017*; *Tarr & Warren, 2002*; *Amirthalingam et al., 2021*; *Laver et al., 2017*). However, the feasibility of VR-therapy for upper extremity mobilization in ICU patients has yet to be explored.

Therefore, the primary aim of this study was to evaluate the feasibility of VR-therapy using a VR-headset during ICU- and subsequent general ward admission. The secondary aim was to observe mobility and handgrip strength at the start of VR-therapy and after four weeks of training or at hospital discharge. We hypothesized that using a dedicated VR-game is feasible for early ICU mobilization, reflecting in session efficiency, adherence and patient satisfaction.

## MATERIALS & METHODS

### Study design

A healthcare innovation feasibility study was performed with a pre-post design. A local medical ethics committee labeled this study as a non-Medical Research Involving Humans Act study (Dutch: Wet medisch-wetenschappelijk onderzoek met mensen, WMO), due to its non-incriminating nature (Regionale Toetsingscommmissie Patiëntgebonden Onderzoek, Leeuwarden, The Netherlands; nWMO-number: nWMO 20210056). Nevertheless, written informed consent for study participation and data collection was obtained. Portions of this text were previously published as part of a preprint (https://pure.rug.nl/ws/files/582211383/Complete_thesis.pdf).

### Population

Patients were recruited from March 2022 through May 2022 at the ICU of the Medical Center Leeuwarden, a tertiary teaching hospital in the Netherlands. To reduce potential selection bias, all eligible patients were consecutively screened by an independent researcher. Inclusion criteria were: ≥18 years old, mechanically ventilated for ≥48 h in the ICU, and capable to provide written informed consent based on assessment by clinical staff. Patients were excluded in case of an active delirium, indicated by an ICU-nurse, clinician, or Confusion Assessment Method for ICU ≥ 1 (*Ely et al., 2001*), and/or if they did not understand Dutch. All patients gave written informed consent for data collection prior to participation.

### Data collection procedures

VR-therapy was offered as a complement to standard daily physical therapy and early ICU mobilization. A VR-headset, the Oculus Quest 2® (Meta Technologies, LLC), was used for VR-therapy. To ensure VR-therapy was suitable for recovering ICU-patients, a dedicated prototype game was developed using participatory design sessions with relevant stakeholders. The design process consisted of the several sessions following the double diamond model (*Ferreira et al., 2015*):

(1) Identify phase: a brainstorm session was conducted with our local post-ICU clinic team, a physical therapist, two dedicated researchers and two game developers to identify relevant problems within early ICU mobilization.

(2) Define phase: combining previous research with clinical practice in our local ICU, a need for upper-extremity specific mobilization practices and a high occurrence of training barriers were defined as key issues in a separate session with two intensivists, a physical therapist, two dedicated researchers and two game developers.

(3) Develop phase: three iterative sessions were held with two former ICU patients and one of their informal caregivers, our local post-ICU clinic team, a physical therapist, two dedicated researchers and two game developers to brainstorm on a possible solution. In addition, both hardware and software were tested using existing rehabilitation games.

(4) Deliver phase: a prototype puzzle game using immersive VR-headsets was developed and tested by the development team. In-hospital experimentation with the VR-headset was conducted to ensure the game would be playable in the ICU and general ward setting without interference of other technological devices present. Finally, this feasibility study was conducted.

These steps resulted in a VR-game in which patients were instructed to complete puzzles with increasing levels of difficulty to train upper extremity functionality (Fig. 1). The puzzles were made on a table-like surface in a virtual home environment, while the hand movements were tracked using inside-out tracking and computer vision techniques, using four outward-facing cameras on the front of the VR-headset, and displayed in the VR-environment. The visual elements in the virtual home environment were designed previously as part of an intervention for loneliness in older adults by 8D Games in collaboration with *Veldmeijer et al. (2020)*.

Based on local physical mobilization protocol and previously conducted studies on VR-therapy for ICU-patients, VR-therapy was offered three times a week for 20 min with a maximum of 4 weeks or until hospital discharge, which will be referred to as the follow-up visit (*Kanschik et al., 2023*). VR-therapy was offered in the ICU and the general hospital ward.

## Safety measures

To ensure the safety of participating patients, several precautionary measures were applied. Although previous studies have identified immersive VR to be safe in patients with active delirium, patients were only included in this study once they were able to provide written informed consent (*Kanschik et al., 2023*). During the VR-therapy session, a trained researcher was present to support the patient and monitor for adverse events. The researcher provided the patient with a brief introduction to the software and then assisted in mounting the VR-headset and selecting the level of difficulty, the number of puzzle pieces, and the use of left and/or right hand. To reduce the risk of falling, patients performed VR-therapy in a seated position in bed or a chair. To ensure that all patients received the same type and complexity of training, every VR-therapy session was personalized to the patient's capacities using a standardized protocol with a step-by-step increase of the difficulty level. Any reported discomfort or complaints were assessed and added to the case report form by the researcher who adjusted the training accordingly. Patients could stop training at any moment without any consequences to their participation in the study or their standard care rehabilitation.

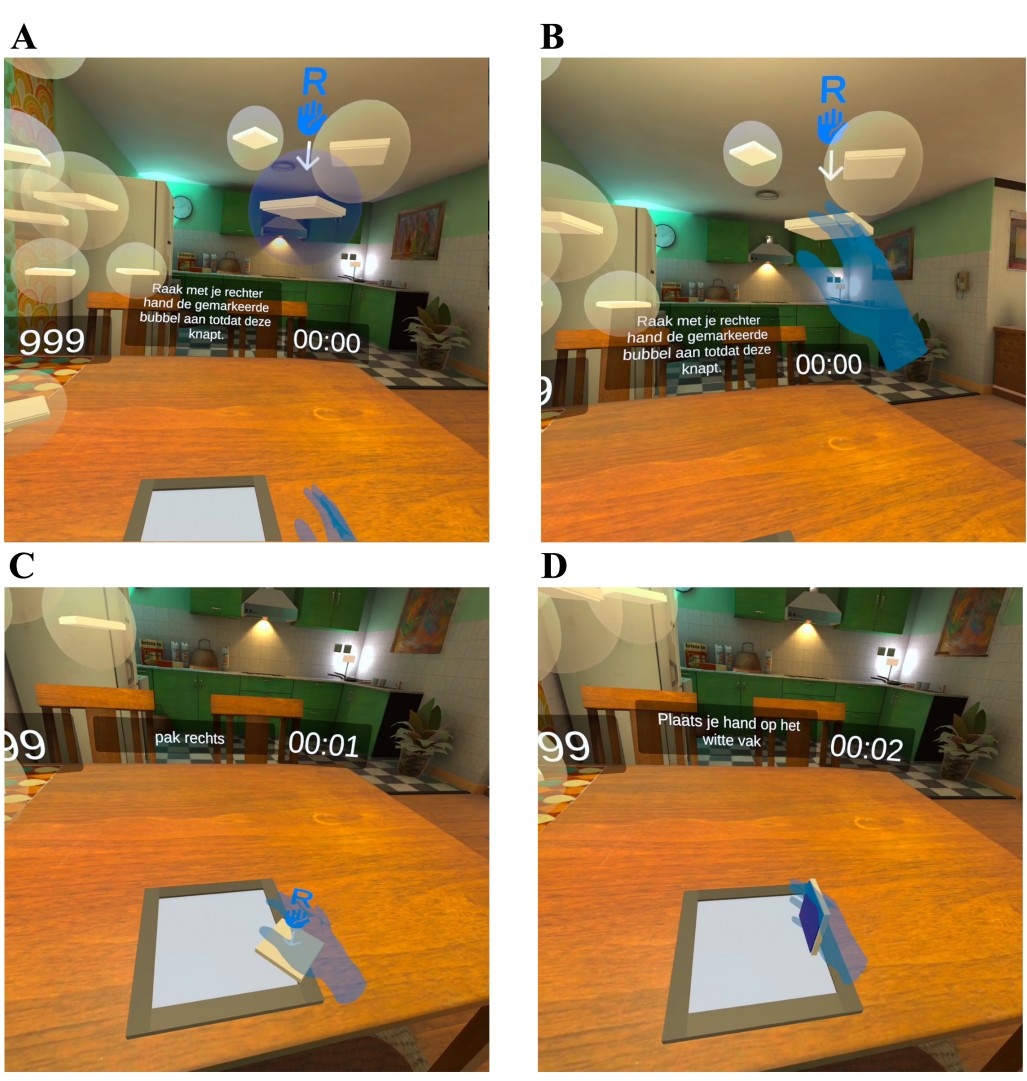

**Figure 1** **Screenshot from VR-game.** (A) One of the bubbles lights up; (B) by touching a bubble, it bursts, and the puzzle piece falls on the table; (C) patients navigate their hand to a fallen puzzle piece; (D) patients can flip puzzle pieces by turning their hand over and put down a puzzle piece by moving their hand down towards the table.

## Outcome measurement tools

Primary feasibility outcomes were the number and duration of VR-therapy sessions, actual time spent using the VR-game, session efficiency (the actual time spent using the VR-game/duration of VR-therapy session*100%), and adherence to VR-therapy. Feasibility outcomes were presented per patient and per VR-therapy session to explore changes over time. During each session, the number and nature of adverse events (*e.g.*, dizziness, pain) were monitored as well as fatigue using the Borg Rating of Perceived Exertion Scale (Borg-RPE) (6–20). The BORG-RPE is a valid and reliable tool to measure exertion and fatigue, with higher scores indicating more fatigue (*Colberg, Swain & Vinik, 2003*). Patient satisfaction levels were measured after each session using a Visual Analogue

Scale (VAS) (0–100), indicating high satisfaction with a higher score (*Voutilainen et al., 2016*). Additionally, self-reported probability of using the VR-game in a home situation using a VAS (0–100), hand-grip strength by a handheld dynamometer (*Baldwin, Paratz & Bersten, 2013*), and the MMI (*Sommers et al., 2016*) were evaluated before the start of VR-therapy and at the follow-up visit, where higher scores indicated more satisfaction, better grip strength and higher mobility. MMI consisted of 15 mobility items (bed, chair, static balance, walking, and dynamic balance items) and ranges from 0 to 100, whereas 0 represents poor mobility and 100 independent mobility. Both the hand-grip strength and the MMI are feasible and reliable tools to measure physical functioning in ICU patients and are part of the standard care protocol of the study hospital (*Beumeler et al., 2020*).

## Statistical analyses

All primary and secondary outcomes were prespecified in the study protocol and reported in this paper to avoid selective reporting bias. Quantitative data were presented as categorical and continuous variables and qualitative data from patient and trainer experiences were described as in-text quotes. Descriptive statistics were used to obtain a detailed picture of the data. Due to the small sample size, non-parametric Wilcoxon signed-rank tests were used to assess differences in self-reported probability of using VR-game in a home situation, MMI, and hand-grip strength between the start of VR-therapy and at the follow-up visit. Analyses were conducted using SPSS Statistics 24 software (IBM, Irvine, CA, USA) with $p < 0.05$ considered to be statistically significant.

## RESULTS

Of twelve eligible patients, ten patients gave permission to participate in this study. Nine patients completed the training. Patient characteristics are shown in Table 1. The median age was 71 [63–79] and 70% identified as male. Three patients had been diagnosed with chronic obstructive pulmonary disease and one with diabetes type 2 prior to ICU admission.

Patients were able to complete informed consent and start VR-therapy after a median of five days after ICU-admission. Patients participated in three VR-therapy sessions per week with a median session duration of 20 min and actual VR-gaming time ranging from 3 to 22 min (Table 2). The remaining session time was used for preparation, giving an introduction to the software, helping the patient to put on the VR-headset, selecting the game settings, resting if needed, interruption by other healthcare providers, and/or restarting VR-headset in case of technical difficulties. This resulted in session efficiencies ranging from 25% to 93%. To illustrate, patients 3 and 10 were very weak and therefore needed more support with the VR headset and could only sustain VR-therapy for a short time reflecting in lower session efficiency. On the other hand, patient 7 really enjoyed VR-therapy and therefore trained with extremely light to very light activity levels to last longer. Patients rated overall satisfaction and fatigue level of 80/100 and 11/20, respectively, indicating high satisfaction and moderate exertion levels. Reasons for non-adherence to VR-therapy were: tiredness (four (11%)), patient was unable to sit up properly (one (3%)),

**Table 1  Patient demographics and ICU characteristics.**

|  | ICU-patients ($n = 10$) |
|---|---|
| Age (years) | 71 [63–79] |
| Male | 7 (70%) |
| BMI (kg/m$^2$) | 27.1 [22.5–29.6] |
| APACHE-III score[a] | 74 [66–104] |
| Frailty score[b, c] | 2 [2–3] |
| Admission type |  |
| Medical | 6 (60%) |
| Elective surgery | 1 (10%) |
| Acute surgery | 3 (30%) |
| Cardiopulmonary resuscitation | 3 (30%) |
| Sepsis | 1 (10%) |
| Medical comorbidities | 4 (40%) |
| Length of stay ICU prior to inclusion (days) | 5 [4–10] |
| Length of stay ICU (days) | 6 [4–9] |
| Length of stay hospital (days) | 12 [10–19] |
| Mechanical ventilation (days) | 3 [3–7] |

**Notes.**

Data are presented as median [IQR] or number (%).

ICU, Intensive Care Unit; APACHE, Acute Physiology and Chronic Health Evaluation.

[a] Ranges from 0 to 299, with higher values representing a worse prognosis (*Voutilainen et al., 2016*).

[b] Ranges from 1 (very fit) to 9 (terminally ill) (*Baldwin, Paratz & Bersten, 2013*).

[c] Missing for five patients.

no motivation (four (11%)), patient saw no added value of VR-therapy (six (17%)), or technical difficulties (one (3%)).

No serious adverse events were experienced by patients or observed the trained researcher. Two patients experienced pain due to fractured ribs and sternum and were unable to play a higher level. Another patient reported some dizziness after VR-therapy. In general, patients experienced VR-therapy sessions as a "fun activity", "special experience", and "fun and at the same time effective activity during the long hospital days". With more consecutive VR-therapy sessions, the session duration, VR-gaming duration, session efficiency, satisfaction level, and fatigue level increased (Table 3). Adherence levels decreased as some patients experienced the game as too easy or repetitive after multiple sessions.

The median self-reported probability of using the VR-game in a home situation displayed an absolute increase, but this was not significant (Table 4). MMI scores significantly increased over time (45 [28–70] to 78 [26–88], $p = 0.005$), indicating better balance and mobility (Fig. 2A). No significant differences in absolute and relative hand-grip strengths (Fig. 2B) were found between the start and end of the training period. We refer to Fig. A1 for data on individual differences in hand-grip strength between these time points.

## DISCUSSION

This was the first study evaluating the feasibility of a co-created, fully immersive VR-game for early upper extremity mobilization in the ICU, providing tailored therapy by adjusting
**Table 2  Summary of feasibility outcome measures, satisfaction and fatigue levels per patient.**

| ID | Nr of research visits | Nr of sessions completed | VR-therapy session duration (min.) | VR-gaming duration (min.) | Session efficiency (%) | Satisfaction level (VAS)[a] | Fatigue level (Borg-RPE)[b] | Adh. (%) |
|---|---|---|---|---|---|---|---|---|
| 1 | 4 | 3 | 25 (20–26) | 19 (19–19) | 74 (71–93) | 90 (75–100) | 13 (13–13) | 75 |
| 2 | 3 | 2 | 20 (9–31) | 8 (6-9)[c] | 49 (28–71) | 75 (60–90) | 11 (10–11) | 67 |
| 3 | 2 | 1 | 10 (10–10) | 3 (3–3) | 25 (25–25) | 75 (75–75) | 13 (13–13) | 50 |
| 4 | 3 | 2 | 33 (32–33) | 17 (15–19) | 52 (48–57) | 85 (70–100) | 11 (11–11) | 67 |
| 5[c] | 3 | 0 | – | – | – | – | – | 0 |
| 6 | 3 | 3 | 20 (15–20) | 7 (6–7)[d] | 39 (35–43)[d] | 80 (80–100) | 11 (9–11) | 100 |
| 7 | 3 | 3 | 32 (25–40) | 18 (15–22) | 69 (37–73) | 90 (80–90) | 9 (7–9) | 100 |
| 8 | 3 | 2 | 19 (18–20) | 11 (10–13) | 59 (53–65) | 63 (50–75) | 11 (9–13) | 67 |
| 9 | 5 | 3 | 20 (12–30) | 10 (8–17) | 64 (34–83) | 80 (70–100) | 7 (7–11) | 60 |
| 10 | 7 | 1 | 20 (20–20) | 7 (7–7) | 33 (33–33) | 75 (75–75) | 13 (13–13) | 14 |
| | All subjects | | 20 (9–40) | 13 (3–22) | 57 (25–93) | 80 (50–100) | 11 (7–13) | 60 |

Notes.
  Data are presented as median (min.-max.).
  Adh, Adherance; ID, patient identification; VR, virtual reality; VAS, visual analogue scale; Borg-RPE, Borg Rating of Perceived Exertion scale.
  [a]Ranging from 0 to 100.
  [b]Ranging from 6 to 20.
  [c]Patient 5 was too tired and short of breath to participate in VR-therapy sessions.
  [d]Missing for one VR-therapy session.

**Table 3  Summary of feasibility outcome measures, satisfaction and fatigue per VR-therapy session.**

| VR-therapy session | VR-therapy session duration (min.) | VR-gaming duration (min.) | Session efficiency (%) | Satisfaction level (VAS)[a] | Fatigue (Borg-RPE)[b] | Ad-herence (%) |
|---|---|---|---|---|---|---|
| 1 | 20 (9–32) | 8 (3–18) | 45 (25–73) | 78 (50–100) | 11 (7–13) | 80 |
| 2 | 26 (18–33) | 18 (9–22)[c] | 63 (28–83)[c] | 80 (70–90) | 11 (9–13) | 70 |
| 3 | 20 (12–40) | 11 (7–19) | 50 (35–93) | 85 (70–90) | 9 (7–13) | 33 |
| 4 | 25 (25–25) | 19 (19–19) | 74 (74–74) | 100 (100–100) | 13(13–13) | 33 |

Notes.
  Data are presented as median (min.-max.).
  VR, virtual reality; VAS, visual analogue scale; Borg-RPE, Borg Rating of Perceived Exertion scale.
  [a]Ranging from 0 to 100.
  [b]Ranging from 6 to 20.
  [c]Missing for one patient.

the level of difficulty, number of puzzle pieces, and use of the left and/or right hand. In this study, we aimed to evaluate the feasibility of VR-therapy during ICU- and subsequent general ward admission. In addition, we explored parameters of physical functioning before and after the start of VR-therapy. Consistent with our hypothesis, it was feasible to offer VR-therapy three times a week for 20 min in addition to standard daily physical therapy in patients with critical illness. No serious adverse events were reported by patients or observed by a trained researcher. Patients reported a high satisfaction level and reported a 78% probability of using the game in their home setting.

This study highlights the feasibility of using immersive VR-therapy for ICU patients when they are in a seated position, either in bed or in a chair. A study by *Parke,*

**Table 4 Difference of probability of using game in home situation, hand-grip strength, and MMI between the start of VR-therapy and the follow-up visit.**

| | Start VR-therapy | Follow-up visit | *p*-value |
|---|---|---|---|
| VAS-score | | | |
|     Probability of using game in home situation | 45 [28–70] | 78 [26-88][a] | 0.066 |
| MMI | 26 [24–44] | 57 [41–85] | 0.005[*] |
| Absolute hand-grip strength[b] | | | |
|     Right hand (kg) | 23.8 [11.9–35.4] | 31.0 [12.7–39.5] | 0.386 |
|     Left hand (kg) | 25.2 [10.5–29.1] | 25.2 [17.3–34.6] | 0.386 |
| Relative hand-grip strength | | | |
|     Right hand (%) | 70.0 [52.2–91.6] | 88.8 [50.4–106] | 0.114 |
|     Left hand (%) | 74.3 [56.7–105.6] | 92.8 [70.0–121] | 0.074 |

**Notes.**
Data are presented as median [IQR].
ICU, Intensive Care Unit; VAS, visual analogue scale; MMI, the Morton Mobility Index.
[*]Significantly different ($p < 0.05$) with Wilcoxon signed rank test.
[a]Missing for one patient.
[b]The right hand was dominant for nine patients and the left hand for one patient.

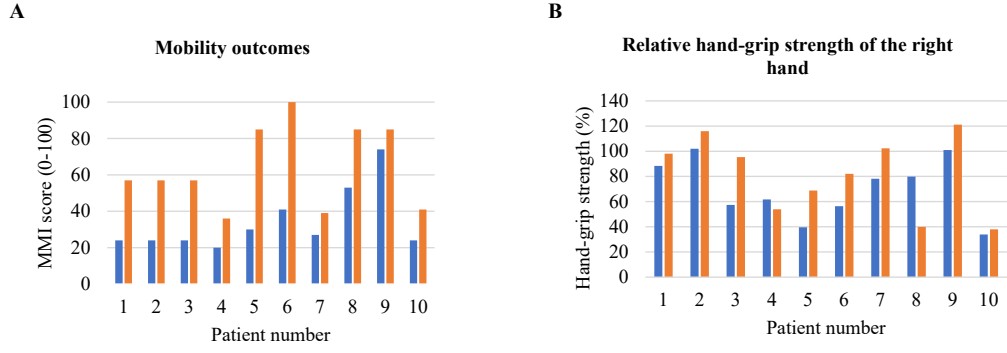

**Figure 2 Individual differences between pre- and post-test.** (A) MMI score; (B) relative hand-grip strength of right hand. Abbreviations: MMI = the Morton Mobility Index.

*Hough & Bunnell (2020)* found similar results using Virtual Therapy Environments for early ICU mobilization, including multiple virtual platforms using the Xbox Kinect Jintronix© software. This was a feasible and enjoyable way to practice range of motion and strength. Other virtual rehabilitation platforms, like the Nintendo Wii™, were successfully applied in the ICU setting (*Gomes, Schujmann & Fu, 2019*). In addition, *Norouzi-Gheidari et al. (2019)* concluded that VR-therapy was feasible in stroke patients with a session efficiency of 49%, which is slightly lower than our session efficiency of 57%. Although our session efficiency improved over time, there was a high variety between patients emphasizing the need for personalized support when using digital tools in recovery.

In general, patients were satisfied with VR-therapy and found it an enjoyable and fun addition to standard care mobilization. Positive user experiences are often reported in feasibility studies regarding VR interventions and highlight the potential of these tools for

clinical practice (*Kanschik et al., 2023*). Although there is limited evidence on the use of VR-exergames by ICU survivors during ICU stay and in the post-acute phase, in stroke patients adherence to a home-based VR-therapy was good (*Jonsdottir et al., 2018*) and the efficacy seems comparable with clinic-based VR-therapy (*Schröder et al., 2019*). In our study, session efficiency and satisfaction level improved after multiple sessions, showing a promising training effect. As recovery of critical illness can be a lengthy process, we explored the hypothetical scenario of using the VR-therapy at in a home setting (*Beumeler et al., 2022*). Our results show that the self-reported probability of using the VR-game at home increased from the start of VR-therapy to the follow-up visit. In line with this, complementing care with applications of ehealth, serious gaming, and remote care may ensure continuity in rehabilitation.

Our predetermined goal to train 20 min was generally not achieved, which highlights the need for more research on optimal training duration of VR-therapy and other exergames in ICU patients. It is unclear what the ideal duration of training is using VR in ICU rehabilitation, as the area is relatively unexplored. In a systematic review by *Kanschik et al. (2023)*, the applied training duration varied from 3 to 55 min in previously published interventions using VR or augmented reality (AR), with an average training time for interventions targeting ICU patients of 17 min. Using immersive technology has several benefits in managing pain and anxiety in ICU patients, but might also cause strain or motion sickness (*Lundin, Yeap & Menkes, 2023*). We observed that VR-therapy resulted in increased fatigue for some patients, affecting adherence rates. Fatigue has been shown to be a reason for activity cessation and a barrier to adhere to exercise in ICU-patients by others as well (*Parry et al., 2017*; *Parke, Hough & Bunnell, 2020*; *Berney et al., 2012*). Possible ways to mitigate the occurrence of fatigue and strain may include a more personalized training approach, offering shorter sessions which gradually increase in duration over time, alternating VR-therapy with relaxing activities or providing the opportunity for patients to practice during the evening, when other care activities are generally limited. Simultaneously, VR-therapy was not challenging enough to stay motivated for 20 min for other patients. Increasing the range of difficulty of the VR-therapy could improve player enjoyment (*Sweetser & Wyeth, 2005*), which should be considered in the further development of VR-therapy.

Although the main focus of this study was the evaluation of feasibility of VR-therapy, changes in physical parameters were explored. Patients reported exertion levels of 7/20 to 13/20 after VR-therapy, indicating that our VR-therapy led to very light to somewhat hard activity levels. Training intensities corresponding to a Borg-RPE range 11–13 are recommended in sedentary, less fit, and untrained individuals, as well as patients with cardiovascular diseases (*Conijn et al., 2024*). This suggests that our VR-therapy met the recommended training intensities for most ICU-patients. Furthermore, balance and mobility significantly improved from the start of VR-therapy to the follow-up visit. Mobility scores at the end of the training period in this study were higher than previously reported in ICU-patients at ICU-discharge (*Sommers et al., 2016*; *Da Silva et al., 2020*), but comparable to mobility scores measured at hospital discharge (*Baldwin et al., 2020*). This

improvement might be attributed to the addition of VR-therapy to standard care, which should be investigated further in future, controlled studies.

Despite the promising results of this study, there are some limitations to take into consideration impacting their generalizability. As a feasibility study, our study design did not include a control group, limiting our ability to assess the efficacy compared to standard daily early mobilization. In addition, the small sample size limited the ability to detect clinically relevant differences. However, in a heterogeneous, severely ill population like this, it was essential to first demonstrate the safety and feasibility of VR-therapy using a fully immersive headset before moving on to larger trials. Nevertheless, this study also had several strengths. The VR-exergame was co-created with patients, relatives, healthcare workers and developers, which ensured that it was tailored to the needs of the involved stakeholders. Also, we demonstrated that it is feasible to initiate VR-therapy in the early stages of hospital admission in a severely ill population. Importantly, we did not observe serious adverse events, like commonly reported motion sickness, during the study.

## CONCLUSION

In conclusion, VR-therapy using a co-created, fully immersive VR-game for early upper extremity mobilization is feasible in the ICU and on the general hospital ward.

### Recommendation

Future studies should examine whether VR-therapy as a complement to conventional therapy improves physical functioning. Furthermore, personalized VR-therapy at home could be valuable addition to rehabilitation practices for ICU survivors.

## ACKNOWLEDGEMENTS

The authors would like to thank all patients, informal caretakers, and healthcare professionals for participating in the various steps of the user-centered design process. We are grateful to 8D Games for facilitating this process, developing the VR-game, and providing continuous technical assistance.

### Funding

The authors received no funding for this work.

### Competing Interests

The research team collaborated with 8D Games for the development of the VR-game. Development costs were covered by a dedicated innovation fund of the Medical Centre Leeuwarden. VR-headsets were acquired by the Intensive Care Unit of the Medical Centre Leeuwarden.

## Author Contributions

- Mirthe de Vries performed the experiments, analyzed the data, prepared figures and/or tables, authored or reviewed drafts of the article, and approved the final draft.
- Lise F.E. Beumeler conceived and designed the experiments, performed the experiments, analyzed the data, prepared figures and/or tables, authored or reviewed drafts of the article, and approved the final draft.
- Johan van der Meulen conceived and designed the experiments, authored or reviewed drafts of the article, and approved the final draft.
- Carina Bethlehem conceived and designed the experiments, authored or reviewed drafts of the article, and approved the final draft.
- Rob den Otter conceived and designed the experiments, authored or reviewed drafts of the article, and approved the final draft.
- E. Christiaan Boerma conceived and designed the experiments, authored or reviewed drafts of the article, and approved the final draft.

## Human Ethics

The following information was supplied relating to ethical approvals (i.e., approving body and any reference numbers):

A local medical ethics committee labeled this study as a non-Medical Research Involving Humans Act study (Dutch: Wet medisch-wetenschappelijk onderzoek met mensen, WMO), due to its non-incriminating nature (Regionale Toetsingscommmissie Patiëntgebonden Onderzoek, Leeuwarden, The Netherlands; nWMO-number: nWMO 20210056).

## Data Availability

The raw data are available in the Supplementary File.

## Supplemental Information

Supplemental information for this article can be found online at http://dx.doi.org/10.7717/peerj.18461#supplemental-information.

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
