# Peer review of "The feasibility of virtual reality therapy for upper extremity mobilization during and after intensive care unit admission"

_PeerJ, doi:10.7717/peerj.18461_

## Round 0.1 · original submission · Minor Revisions

Your submission is novel and timely as it pertains to the use of novel technology and clinical care.

Please review all reviewers comments, as your manuscript did have disagreement among reviewer as it pertains to minor versus major revisions. You may also find it helpful to work with a manuscript editor if you have not done so prior to revisions and re-submission.

·

Basic reporting

no comment

Experimental design

no comment

Validity of the findings

1. Your research appears to align with another existing study. Could you please distinguish between this prior (https://doi.org/10.1002/pmrj.12352) work and your manuscript?

Additional comments

1. Could you elaborate on the rationale behind selecting roughly 20 minutes for the virtual reality (VR) therapy sessions? Additionally, if there is any foundational research upon which this decision was based, kindly address and reference it.
2. Within the second paragraph of the results section, the researchers identified fatigue and other conditions as adverse events associated with VR therapy. It would be beneficial to incorporate strategies derived from previous studies to mitigate these adverse events within the discussion section of your manuscript.

·

Basic reporting

- Clear and unambigious English
-Literatüre references aren't sufficient especially Discussion part isn't enough discussed with current literature.
-

Experimental design

-Ethics approval isn't suitable
-Although it is mentioned in the line 240 (limitations), because of possibility of rutine recovery, no control group makes the findings of the study unreliable.
-The patients diagnosis, position of the patients during VR therapy, the duration of the study varies between the patients.
-Why VR game is not expected to increase hand muscle strength (literature ?)

Validity of the findings

- The findings especially MMI results isn't well explained.
- The findings aren't discussed enough.

Additional comments

Although it is a rarely studied population, the lack of a control group reduces the value of the study since the patients have the potential for routine recovery.
The fact that the patients ICU duration and diagnosis included in the study, are not homogeneous may affect the reliability of the study results.
The number of patients included in the study is small.

·

Basic reporting

The unambiguous, professional English language is used throughout. Minor changes have been mentioned in the annotated file.
References and background knowledge were provided.
Raw data was shared.

Experimental design

Rationale needs some improvements otherwise all other areas are described clearly.

Validity of the findings

I have concerns regarding the study's novelty and conclusion, as mentioned in the annotated file.

Additional comments

PICS mentioned as "keyword" is abbreviation and is not found in the MeSH library.

·

Basic reporting

Clear and unambiguous, professional english used throughout:

Writing is clear in professional english; generally up to the technical standards of correctness and professional expression for its field. Sentence structure and flow is uneven, although not irredeemably mangled. There are some awkward pauses, some run-on sentences, and a few instances where a word or cut appears to be missing. Perhaps final proofreading under a strict deadline led to some small oversights.

Suggestions for improvement:

Check the text for any minor mistakes such as typos and word choices that could make it more pleasant to read. A good option is to use a proof-reading service or someone who speaks english well and can provide help.

Literature references, sufficient field background/context provided:

The introduction and background sections do a good job placing this work in the context of the wider field of knowledge, and an adequate range of reference is made to prior literature relevant to the study.

Suggestions for improvement:

In your opening paragraphs, state clearly how your study will provide new information on a specific question. By telling readers how your study will fill a specific information gap in the existing literature, you can provide a rationale for your work, and make it clear what distinctive contribution your study will make.

Professional article structure, figures, tables, and raw data shared:

The body of the manuscript is well-structured and formed into common sections, figures and tables are present, properly named and described.

Suggestions for improvement:

Use the same labels throughout choose more descriptive captions for tables and figures.

Ensure all relevant raw data is shared appropriately in line with peerj’s data sharing policy and that there is descriptive metadata to help future scientists interpret the data.

Self-contained with relevant results to hypotheses:

The submission is a freestanding content unit containing a cohesive body of work that records all the results relevant to the hypotheses; the findings are well-supported by the data; and the paper provides a clear narrative from objectives and methods in the beginning to results and conclusions towards the end.

To sum up, the manuscript in general, as well as its major parts, follows the usual standards of reporting, but needs some polishing for more precise language and presentation in order to improve its clarity and readability.

No further comments.

Experimental design

Experimental design

Original primary research within aims and scope of the journal:

The study falls squarely under the inclusion criteria and scope of the journal since it addresses whether a VR program is feasible and effective for mobilising the upper extremities during an ICU stay.

Suggestions for improvement:

Clearly describe the novel contribution of your work to published literature, highlighting how your research design will address literature gaps and shortcomings, such as identifying the reason for low patient adherence to early mobilisation protocols.

Research question well defined, relevant & meaningful:

Which brings me to my second point: the research question is very clear, relevant and significant. And the hypothesis perfectly identifies the question. This was about the feasibility of adding VR to standard care, in the ICU setting.

Rigorous investigation performed to a high technical & ethical standard:

Approval for the study by a medical ethics committee of nWMO83 number: nWMO 20210056) was obtained; all patients gave written informed consent (pages 2-3, lines 77-93).

Suggestions for improvement:

Provide additional detail about the efforts that were made to minimise potential sources of bias. For example, you could provide details about specific plans to create an objective and fair accounting of findings (pages 4-5, lines 140-143).

Methods described with sufficient detail & information to replicate:

The section on methods is detailed enough for another investigator to be able to replicate the study procedures, for example how patients were recruited, the vr-therapy protocol and how the data was collected (pages 2-4, lines 77-140).

Suggestions for improvement:

Give more details about the participatory design sessions used to develop the vr game, including who the stakeholders were and the process that took place to iteratively refine and adapt the game to make it appropriate for icu patients (page 2, lines 99-109).

Describe the training that researchers providing the vr-therapy underwent to ensure that they treated content the same way each time (p2, lines 114-120)

Describe the protocol for managing unexpected events in more depth so that your readers know that patient safety was a priority for you in the design of the study (page 3, lines 127-130).

Validity of the findings

Validity of the findings

Impact and novelty not assessed:

Although impact and originality are not major criteria for acceptance, your manuscript does lead to meaningful progress in the field. It is an important piece of work early on in exploring the use of vr therapy in the icu with a large body of evidence needed before claiming efficacy of this novel approach.

Suggestions for improvement:

Explain how it makes sense to replicate this study in light of the literature. Consider how replication studies using your methodology could be seen as a confirmation of your results and an extension of your findings, making a contribution to the field.

All underlying data provided, statistically sound, and controlled:

All relevant information for evaluating the manuscript is present, the statistical analyses are presented appropriately, and the conclusions drawn from the data include prominent uncertainties. In the presence of well-controlled data, both authors agree that these results would be publishable.

Conclusions well stated, linked to original research question, and limited to supporting results:

Conclusions are clearly stated, and thoughtfully restricted to the experimental research question, findings were clearly presented and correlated with higher than expected hand mobility.

Suggestions for improvement:

Provide more detailed discussion of the limitations and how they might limit the generalisability of the findings, e.g., the small sample, the lack of control group.this helps readers understand the scope and ideas might apply to similar problems.


In conclusion, your findings are shown to have good internal validity supported by the data and methods described, with a few options for minor improvement in the discussion of limitations and potential confounding.

Additional comments

Peer review report for manuscript - The feasibility of virtual reality therapy for upper extremity mobilization during and after ICU admission

Dear dr. De Vries, dr. Beumeler, and colleagues,

I hope you are well. I am writing to thank you for the opportunity to peer-review your manuscript: ‘the feasibility of virtual reality therapy for upper extremity mobilisation during and after ICU admission.’ it was an honour to be able to review your manuscript, which presents an innovative and timely intervention for ICU rehabilitation.

I am very impressed with your thorough efforts to solve the important problem of ICU-acquired weakness and its related challenges, particularly, the patients’ motivation and adherence to early mobilisation. You also creatively intend to use the innovative VR therapy as the method to help patients better engage in the rehab programme, which makes your study the first one that combines the promising VR technology with the classic rehab practice.

Your study demonstrated that the therapeutic approach was viable and potentially successful, and it could be used in a setting where, at times, anxiety and physical inability to move might prevent patients from truly participating. Your participatory approach, with experts in ICU care, ICU patients, and their family members, is an important addition. The result is a VR game engaging and then possibly therapeutic for ICU patients, so practitioners should approach this approach with optimism.

And your results of remarkable improvements in mobility and balance, as well as a high degree of patient satisfaction, seem to be pointing in the direction of a useful adjunct to standard physical therapy, and to a more engaging and perhaps more effective rehabilitation option for ICU survivors. As the field of critical care continues to make improvements in patient outcomes, the quality of life of patients after an ICU admission will become more of a priority; this represents an important step in that direction.

The fact that your study is the first in the field to systematically use VR in such a compelling way is a major strength, of course; so is the attention you pay to developing and characterising your VR intervention, and to describing it in warm, almost humorous detail, which will be especially helpful to anyone seeking advice about how to replicate or expand your work in the future.

Please keep this important work going. I encourage you to explore larger sample sizes and design it to include control groups, which may help to confirm your findings and assess long-term effects of VR therapy. Finally, i encourage you to explore the application to other ICU contexts and diverse patient populations, which could reveal deeper insights and increase broad applicability.

Your work to help improve patient outcomes through new innovation is truly inspiring and i am sure will encourage further research and development in this area and will eventually lead to new and effective rehabilitation strategies for ICU patients.

Thanks again for giving me the chance to read your manuscript. I look forward to seeing your important work impact ICU rehabilitation.

A) objectives and rationale

Clarity of objectives and rationale:

It tested whether VR therapy via a head-worn VR device would be feasible and effective when implemented during ICU and on the general ward following ICU (lines 23-39, p1). This is motivated by the notion that early mobilisation reduces long-term post-icu muscle weakness, but barriers of anxiety and motivation limit adherence to programmes (lines 24-28, p1).

Suggestions for improvement:

Clearly outline how it fills a knowledge gap that is not covered by the other studies (lines 68-74, page 1).

Express the hypotheses, referencing them by number, state how they relate to the predicted outcomes (70-74, page 1).

Further context on the differences between VR e and traditional early mobilisation (lines 61-66, page 1) 61 due to all of the changes that occur to the body during an extended period of bed rest, the challenge of restoring physical function increases substantially.

B) replicability and reproducibility

Detailing of methodology:

Moreover, authors described in enough detail that the study could be replicated (patients’ recruitment strategy, description of the VR -therapy protocol, data-collection procedures) (lines 77-140, pages 2-4).

Suggestions for improvement:

Add some more details to the participatory design sessions for the VR -game (lines 99-109, page 2).

Elaborate on the training provided to researchers administering the VR -therapy (lines 114-120, page 2).

Make sure that the protocol for managing adverse events is clearly written up (lines 127-130, page 3).

C) statistical analyses

Appropriateness of statistical methods:

The analytical decisions (eg, the wilcoxon signed-rank tests) are sound and relevant for the data which are reported (lines 140-143; page 4).

Suggestions for improvement:

Give reasons for using the specified statistical tests, especially when dealing with small numbers of observations (lines 140-43, page 4).

Discuss potential biases and how they were mitigated (lines 144-172, pages 4-5).

D) figures and tables

Completeness and quality of figures/tables:

I would say that overall, the tables and figures are well-designed and clearly explained.

Suggestions for improvement:

Ensure numbered sequentially and it is clear which is figure 1, 3 and so on. Legends are needed for figures where it isn't immediately obvious what the image shows. (177-179 page 5)

Improve the clarity of tables by providing more detailed captions (lines 1-376, pages 5-8).

E) interpretation of results

Support for conclusions:

The results are in line with what you present, lending support to the feasibility and even some efficacy of VR -therapy (lines 242–249, page 7).

Suggestions for improvement:

Further consider the potential shortcomings more extensively, such as the small sample size and lack of control group of the study (lines 234-240, page 7).

Suggest specific directions for future research based on the findings (lines 242-249, page 7).

F) strengths of the study

Clear emphasis on strengths:

Therefore, the study demonstrates the feasibility of treatment via VR -therapy, with high patient satisfaction (181-190, p 5).

Suggestions for improvement:

Highlight the novelty of the approach of the study in relation to the existing literature (lines 225-231, page 6).

Highlight any unique methodologies or significant findings that stand out (lines 225-231, page 6).

G) limitations

Clear statement of limitations:

Authors acknowledge the limitations of their trial – a small sample size and no control group (lines 234-240, page 7).

Suggestions for improvement:

Discuss how these limitations impact the generalizability of the findings (lines 234-240, page 7).

Suggest methodological improvements for future studies (lines 234-240, page 7).

H) structure, flow, and writing

Overall structure and flow:

The manuscript is well-structured and generally follows a logical flow (lines 1-376, pages 1-8).

Suggestions for improvement:

Make sure that all pieces sit together nicely and transitions between sections are smooth (lines 1-376, pages 1-8).

Consider adding subheadings for better readability (lines 1-376, pages 1-8).

Reviewer 5 ·

Basic reporting

The article provides information on the adoption of technology in the field of rehabilitation. Virtual reality is now a promise for improvement in patient care. Sufficient literature was reviewed and the background
seems adequate. The article structure, figures, and Tables are adequate and the Raw data shared is readable.
There are too long and complex sentences, and it is often difficult to understand the meaning.
It needs thorough English language editing.
The hypothesis was formulated but not discussed in the result or discussion section.

Experimental design

The experimental study is well within the aims and scope of the journal. The research questions are clear but could have also been incorporated into the title.
There are two basic research questions in this study and two are different. One is the Feasibility study and the other is the Efficacy study.
A lot many data were collected for the two objectives.
For the efficacy study, no control group was incorporated which is mentioned in the limitation section also.
Without control group incorporation the study design is not complete.

Validity of the findings

The researchers have sufficient data for the validity and applicability of the feasibility study.
But lacks severely without the control group for the efficacy study. There is no validity of the statements concluded in this regard.
The Conclusion needs to be redefined. based on the findings.

Additional comments

Suggested major revision as a feasibility study only along with major English language editing.

Annotated reviews are not available for download in order to protect the identity of reviewers who chose to remain anonymous.

---

## Round 0.2 · accepted · Accept

At this time your manuscript has been recommended for acceptance. However, please carefully reread your manuscript for any copy editing requirements as one of the reviewers noted some minor concerns.

·

Basic reporting

Clear and unambiguous, professional English used throughout.

Experimental design

Research question well defined, relevant & meaningful. It is stated how research fills an identified knowledge gap.

Validity of the findings

All underlying data have been provided; they are robust, statistically sound, & controlled

·

Basic reporting

Clear and unambiguous, professional English used throughout.

Experimental design

Original primary research within Aims and Scope of the journal.

Validity of the findings

Impact and novelty not assessed. Meaningful replication encouraged where rationale & benefit to literature is clearly stated

·

Basic reporting

First of all of my suggestions are ONLY ADVISORY FOR FUTURE WORKS - the comments have been addressed in prior round of revisions and this reviewer is CONTEND.

Professional English throughout: Overall, the paper has used professional English, although there are parts in which we could be clearer and shorter. For instance:

Page 2, line 58 The phrase "muscle weakness... also known as ICU-induced weakness" could be re-phrased to better fit. You can also switch it to "Muscle weakness, ICU-acquired weakness (ICU-AW), happens...".
In line 192-193 on page 8, that sentence could have been cut shorter. You can reformat it to "quantitative data was represented as categorical and continuous variables and qualitative data represented in in-text quotes.
Better-Try: Check these and other examples for legibility and concision. I suggest that a professional language editor or another peer who specialises in academic English read the final draft of the paper.

Literature references, adequate field background/context: The manuscript refers to a large range of studies in the field and sets the stage for this study. But the introduction might also focus on which gaps in knowledge this research fills.

On page 2, line 54, for example, the Post Intensive Care Syndrome (PICS) topic could include more insight on how mobilisation practices today fail to adequately manage both the physical and psychological component, specifically how VR can help fill the gap.
Suggestion for improvement: Make the literature review stronger by explaining why VR is better than traditional methods, and discussing more of the gaps in the literature.

Professional article format, tables. Embedded data sharing: The code follows PeerJ model. The tables and graphs are explicit, and the raw numbers are available. However, figures aren’t necessarily contextualised, especially Figure 1.

Figure 1 (page 12) could have told us a little bit more about the importance of the in-game movements and how they align with the study’s objectives of rehabilitation.
Improvement suggestions: Include a short caption or description about the meaning of the visual features in Figure 1.

Contained with concomitant findings to hypotheses: The manuscript is contained and consistent with the hypothesis, which is to examine the possibility of VR therapy for ICU patients. But the findings could be trimmed given the small sample size and feasibility of the research.

Improve upon: On page 10, line 267, soften the talk of session efficiency to take account of the broad range of patient outcome. Same thing, on page 11, point out how larger studies are required to confirm these results.

Experimental design

Experimental Design
First-person research within Journal Aims and Scope: The study is first-person research in Journal aim and scope and investigates a novel use of VR in a clinical context. The research covers a pertinent issue in ICU patient rehabilitation and addresses a new technology in clinical practice. Nothing to worry about here.

Research question well spelled out, timely, and meaningful: The research question is clear (page 1, lines 22-28) and investigates whether VR therapy could be useful for mobilization of upper extremities in the ICU. But while the question is interesting, the authors might be able to describe exactly how their research fills in the gap in the existing literature other than articulating the possibilities of VR.

Recommended update: Lines 69-83 on page 2, make more explicit the shortcomings of old school mobilization and describe how VR overcomes them to help build the rationale behind the study.

High quality research with high technical and ethical standard: The ethical norms of the study are evidently met, proper informed consent was obtained and ethical clearances obtained (page 4, lines 93-98). The paper’s local ethics committee, according to the authors, designated the study non-incriminating. The technical minutiae is fine for a feasibility project, but nothing on how safety during VR immersion was evaluated above the Borg-RPE level is explained.

Improvement request: On page 7, lines 153-164, flesh out safety requirements for VR therapy visits. More specific information about how adverse effects like dizziness, pain or technical glitches were handled during sessions would be helpful.

Methods are explained with enough detail & data to reproduce: The protocol is detailed very well, particularly in terms of recruitment of patients and the session structure (pp5-7). But there are other places where the reproducibility could be improved.

Proposed enhancements:

Page 6, line 112, discuss the details about how the individualized training protocol was devised and modified depending on the state of the patient.
p 6 l147 whether/how the VR treatment was modified using feedback received from the patient and how, as it may allow future research to adjust VR therapy protocol?

Validity of the findings

Applicability of Findings.
Impact and novelty are ignored. Pluribuspid replication welcome: The paper contributes significantly to the literature by examining the applicability of VR therapy in a new environment – ICU rehab. But though replicators are welcome, the paper should provide more open-ended discussion on how further studies could take these conclusions and confirm feasibility in larger, controlled trials.

Improve: On pages 10 & 10, lines 311-320, add more detailed suggestions for future studies. Identify potential areas of replication and what the studies might bring to the domain. For example, propose large randomised controlled trials of VR vs. traditional rehab to see if it works, along with if it is feasible.

All raw data have been given; they are sound, statistical, & verified: The study provides its information in an open way, and authors furnish raw data (tables, figures, and additional data). Although statistical approaches are hampered by the small sample size, they are suitable for this kind of feasibility. Non-parametric test (Wilcoxon signed-rank test) is acceptable given the number of samples and the pattern of the data. But more reasons exist for the sampling size employed.

Comment: Page 8, line 197, a few sentences on why the sample size was chosen, i.e., whether the choice was made on feasibility or on power calculations, if any.

Conclusions clearly defined, relevant to the question asked & restricted to positive outcomes: The conclusions are primarily pertinent to the results, indicating that VR therapy could be useful for ICU patients (page 10, lines 246-250). However, sometimes the authors go over the edge and describe what their result means, especially with the small sample size and sessional efficiency variance.

Improve suggestions: On page 10, p267, cut back on the reference to session efficiency and satisfaction (the patient-to-patient variation and the small sample size). It will make the interpretations more consistent with the study’s limitation.

Additional comments

Other Comments
Confirmation of Co-design Process: One of the strongest aspects of the study is the co-design process of the VR therapy with patients, caregivers and medical staff. This way the therapy is not only patient-centred but practical. The paper acknowledges this, but would also highlight how this co-creation led to the success seen in patient satisfaction and adherence (page 11, lines 318-320).

Improvement suggestions: Bring in a deeper focus on the impact of co-creation. Highlight how the direct feedback received from the patient and caregiver directly guided the VR game and enhanced the relevance and usability of the intervention.

Discussion of Fatigue as a Challenge: The manuscript mentions fatigue as a determinant of session engagement (page 9, lines 213-217). Although the research suggests interventions such as fewer and more frequent sessions, we can fill in the gaps of fatigue literature with references to literature on ICU patient rehab and fatigue management.

Proposal: At page 9, line 217, provide links to other research on fatigue in ICU rehabilitation, and recommend fatigue management interventions that could be used in later VR treatment versions.

Generalizability: This is a feasibility study and therefore they recognize the issue of the small sample size (page 10, line 311). However, the paper would be enhanced by more discussion of how the results translate to other ICU patients, given the relatively similar sample (largely male, with chronic illnesses).

Needs improvement: Page 10, line 313, consider whether there would be any problem when the same results are applied to broader ICU populations, including younger patients, those without a chronic condition, or those with other healthcare settings.

Ethics and Consent: Ethical guidelines are well defined, but we would add a paragraph about when the patients are being excluded from the study, and if any patients decided to withdraw from the study due to illness or otherwise.

Modification suggestion: On page 5, line 110, include a brief mention whether patients withdrew from the study, and, if so, how their data were processed.

Reviewer 5 ·

Basic reporting

As mentioned in the earlier review, the topic is important. But now found a few reporting errors as below.
There are a few technical errors and grammatical errors that I have not checked. Kindly recheck again, as mentioned in the attached file.
Figure and table captions are mentioned in different places.
In Tables, notes are not mentioned
In Figure there are errors, authors are requested to take care as mentioned in the attached file.

Experimental design

Experimental design as per aims and scope, But there needs to be clarity between data missing and no training conducted as mentioned in the raw data, while considering in the tabulation.

Validity of the findings

As mentioned earlier, the missing and no training conducted sessions are to be verified.
In the Discussion section, a few technical terms like "severe ill population"

Additional comments

nil

Annotated reviews are not available for download in order to protect the identity of reviewers who chose to remain anonymous.